# Laplacian Analysis Meets Dynamics Modelling: Gaussian Splatting for 4D Scene Reconstruction

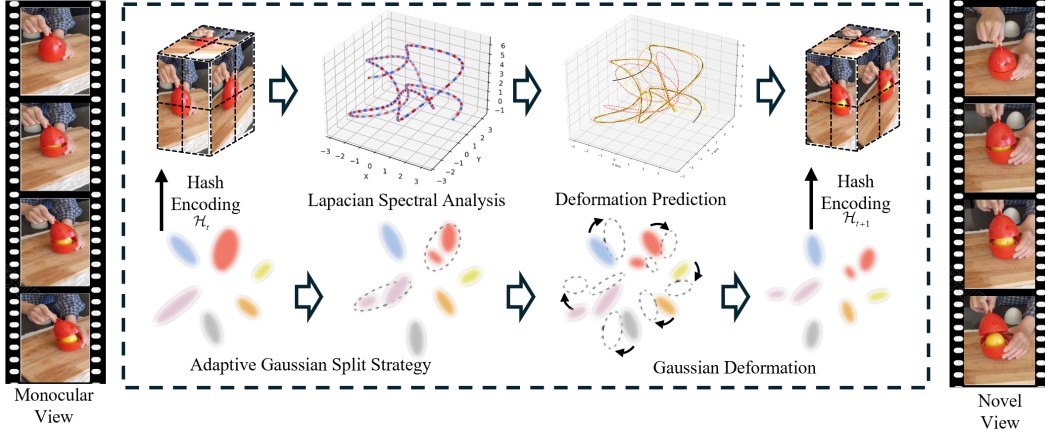

Figure 1: **Overview of our model.** This work integrates Laplacian analysis to model the motion flow more efficiently for 4D Gaussian splatting.

## ABSTRACT

While 3D Gaussian Splatting (3DGS) excels in static scene modeling, its extension to dynamic scenes introduces significant challenges. Existing dynamic 3DGS methods suffer from either over-smoothing due to low-rank decomposition or feature collision from high-dimensional grid sampling. This is because of the inherent spectral conflicts between preserving motion details and maintaining deformation consistency at different frequency. To address these challenges, we propose a novel dynamic 3DGS framework with motion flow extraction. Our approach contains three key innovations: a spectral-aware Laplacian motion flow extraction module which merges Hash encoding and Laplacian analysis for flexible frequency motion control, an enhanced Gaussian dynamics attribute that compensates for photometric distortions caused by geometric deformation, and an adaptive Gaussian split strategy guided by KDTree-based primitive control to efficiently query and optimize dynamic areas. Through extensive experiments, our method demonstrates state-of-the-art performance in reconstructing complex dynamic scenes, achieving better reconstruction fidelity and real-time rendering.

## 1 INTRODUCTION

Dynamic scene reconstruction from monocular videos presents a critical challenge in computer vision, demanding precise modeling of both persistent geometric structures and transient deformations Cai et al. (2022); Du et al. (2021); Fang et al. (2022); Kratimenos et al. (2024); Li et al. (2022). Unlike static environments, dynamic scenes exhibit heterogeneous motion patterns - rigid components maintain temporal consistency while deformable regions require high-frequency trajectory modeling Xu et al. (2024); Duan et al. (2024); Bae et al. (2024). This inherent complexity creates dual challenges: preserving spatial coherence across time-varying geometries and capturing transient deformation details without over-smoothing artifacts.

While Neural Radiance Fields (NeRF) Mildenhall et al. (2021); Chan et al. (2022); Yang et al. (2022); Park et al. (2021b); Martin-Brualla et al. (2021) revolutionized static scene modeling through continuous volumetric integration, their dynamic extensions Choe et al. (2023); Gao et al. (2021); Liang et al. (2023b); Liu et al. (2023); Wang et al. (2023); Barron et al. (2021); Fridovich-Keil et al. (2023) reveal critical limitations in handling temporal discontinuities, particularly the conflicting requirements for spatial fidelity versus temporal coherence arising from uniform spectrum allocation. Although explicit representations Barron et al. (2023); Cao & Johnson (2023); Tancik et al. (2022) improve efficiency through 4D spacetime factorization, their low-rank decomposition induces feature collision in overlapping regions. Recent 3D Gaussian Splatting (3DGS) Kerbl et al. (2023); Duisterhof et al. (2023); Yang et al. (2023); Liang et al. (2023a); Lin et al. (2024) has achieved impressive effects for static environments, where discrete volumetric primitives enable both photorealistic rendering and computationally efficient optimization through differentiable rasterization Shao et al. (2023); Wu et al. (2024); Lu et al. (2024); Luiten et al. (2024); Kratimenos et al. (2024).

However, their direct extension to dynamic scenarios faces three fundamental limitations (detailed theoretical analysis can be found in Appendix. E): 1) existing deformable methods suffer from either over-smoothing due to low-rank decomposition or feature collision from high-dimensional grid sampling, 2) previous Gaussian-based methods use a fixed threshold during Gaussian split stage which ignore adaptive split adjustment, and 3) persistent per-gaussian dynamics caused by intricate deformation are often neglected in current pipelines.

To address the challenges above, our key insight lies in addressing the anisotropic spatio-temporal sampling nature of dynamic scenes through hybrid explicit-implicit encoding. First, we develop a hybrid spectral-aware Laplacian motion flow extraction module that decouples spatial and temporal features into different frequency motion components, overcoming the feature collision of low-rank assumption while enabling adaptive frequency motion control. Then, we design an enhanced Gaussian dynamics attribute to perform individual Gaussian personalized dynamic optimization and design an adaptive regularization for identifying highly dynamic areas. Besides, we propose an adaptive Gaussian split strategy, which focuses on the optimization trade-off between Gaussian shape and anisotropy in dynamic scenes and an improved KDTree-based clustering algorithm was proposed to efficiently query and optimize dynamic Gaussians.

Our solution rethinks dynamic 3DGS through Laplacian spectral analysis, which provides a hybrid framework for localized frequency analysis. Meanwhile, we focus on the dynamics attribute of each Gaussian and the optimization problem in the derivation process, and propose a novel hybrid explicit-implicit algorithm model. In summary, our contributions are as follows:

- We propose a spectral-aware Laplacian Motion Flow Extraction module combining Hash encoding with Laplacian analysis that decouples different frequency motion trajectories from complex deformation.

- We design an enhanced Gaussian dynamics attribute that identify highly dynamic areas for adaptive split and regularization.

- We design an adaptive Gaussian split strategy that automatically adjusts the primitive density and anisotropy using KDTree-guided spectral analysis.

## 2 RELATED WORK

### 2.1 NERF-BASED DYNAMIC MODELING

The advent of Neural Radiance Fields (NeRF) has significantly transformed the landscape of 3D scene reconstruction, particularly for static environments. However, extending NeRF to effectively model dynamic scenes remains a formidable challenge. Early works, such as D-NeRF Pumarola et al. (2021) and Nerfies Park et al. (2021a), have employed canonical space warping and temporal latent codes to capture motion. Despite their innovative approaches, these methods often exhibit limitations when dealing with rapid or abrupt movements. Moreover, explicit spacetime factorization techniques, such as HexPlane Cao & Johnson (2023), have been proposed to enhance computational efficiency. However, these methods impose restrictive low-rank assumptions that may oversimplify the intricate dynamics present in real-world scenes, particularly in environments characterized by rapid changes. Furthermore, while segmenting scenes into components with distinct attributes has

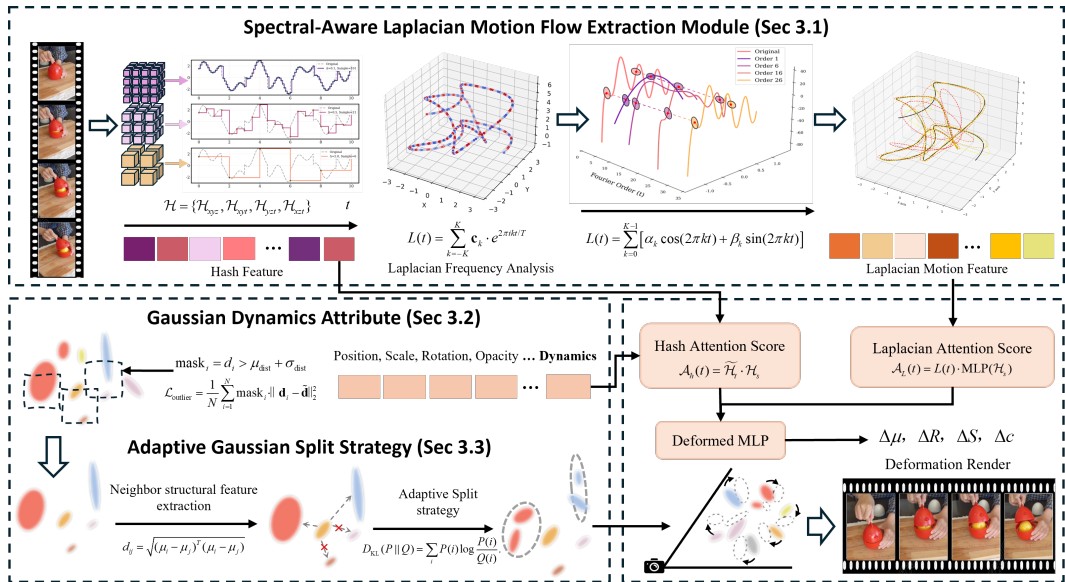

Figure 2: **Framework of our method.** We begin with a spectral-aware laplacian motion flow extraction module for motion trajectory tracking and adaptive frequency analysis. We design a Gaussian dynamic attribute with the spatio-temporal feature by attention mechanism. The whole pipeline benefits from the adaptive dynamic Gaussian split strategy for better performance and efficiency.

been explored to enhance modeling accuracy Gao et al. (2021); Tretschk et al. (2021), the implicit representations based on MLP often suffer from over-smoothing and lengthy training processes.

## 2.2 3DGS-BASED DYNAMIC MODELING

Recent advances in 3D Gaussian Splatting (3DGS) Kerbl et al. (2023); Yu et al. (2024); Huang et al. (2024); Li et al. (2024) have demonstrated remarkable success in static scene reconstruction, prompting extensions to dynamic scenarios. While 4D-GS Wu et al. (2024) employs multi-resolution Hex-Planes with MLPs for deformation modeling, it inherits the fundamental limitation of plane-based methods: the low-rank assumption leads to feature collisions and rendering artifacts in complex motions. Neural deformation fields Yang et al. (2024); Huang et al. (2024) address this through MLPs, but often produce over-smoothed results and struggle with high-frequency details due to insufficient inductive biases. Direct optimization of 4D Gaussians Yang et al. (2023); Duan et al. (2024) offers greater flexibility but introduces optimization challenges including floating artifacts and requires extensive training with additional regularizers. Grid4D Xu et al. (2024) has achieved impressive performance through combining triplane and Hash-coding while it often lacks smoothness and works without an explicit method for modeling dynamic processes. While SplineGS Park et al. (2024) proposes a pipeline that combine 3DGS and spline functions, however, it requires massive priors such as 2D trajactory and depth estimation to maintain performance. These limitations collectively highlight the need for a representation that balances expressiveness with efficient optimization for dynamic 3DGS, particularly in handling complex motions while preserving fine details. Our work addresses these limitations through a novel Laplacian motion flow extraction method that jointly optimizes for physical plausibility, computational efficiency, and multi-scale temporal fidelity.

## 3 METHODOLOGY

In this section, we present our methodology aimed at addressing the challenges of modeling 4D dynamic scenes with high-fidelity spatial details and complex temporal variations. The key innovation lies in a hybrid explicit-implicit representation that combines Laplacian analysis with spectral decomposition to capture spatial features and adaptive temporal dynamics. This framework is structured into three main components: spectral-aware Laplacian motion flow extraction module, enhanced Gaussian dynamic attribute and adaptive Gaussian split strategy. This hybrid approach is

designed to enhance the representation of motion dynamics while maintaining physical consistency across spatial and temporal domains, significantly outperforming existing methods in handling complex motion patterns. The overall pipeline is shown in Fig. 2.

### 3.1 SPECTRAL-AWARE LAPLACIAN MOTION FLOW EXTRACTION MODULE

This section focuses on the challenge of effectively encoding spatial and temporal information to capture the dynamics of motion. We employ a spectral-aware Laplacian motion flow extraction module that decomposes the frequency motion trajectories to accommodate the complexities of 4D spacetime. More theoretical analysis can be found in Appendix. E.

#### 3.1.1 MULTI-SCALE HASH ENCODING

To efficiently encode 4D spacetime information while preserving both spatial and temporal details, we employ a multi-scale hash encoding strategy that extends traditional methods. Inspired by Grid4D Xu et al. (2024), we extend InstantNGP's Hash encoding Müller et al. (2022) to 4D spacetime $(x, y, z, t)$ through anisotropic multi-resolution decomposition.

$$\mathcal{H}^l = \{\mathcal{H}^l_{xyz}, \mathcal{H}^l_{xyt}, \mathcal{H}^l_{yzt}, \mathcal{H}^l_{xzt}\}, \quad l \in \{1, ..., L\} \tag{1}$$

Each level $l$ maintains dimension-specific resolutions computed via geometric progression.

#### 3.1.2 LAPLACIAN ANALYSIS FOR MOTION FLOW PREDICTION

For dynamic scene reconstruction, accurately predicting motion dynamics is paramount. Traditional methods often rely on MLP or linear interpolation, which fails to capture the complex periodic and aperiodic motions present in real-world dynamic scenes. To overcome these limitations, we propose a novel hybrid Laplacian motion flow representation that combines spectral analysis and learnable neural components, allowing for the effective capture of both low and high-frequency motion dynamics. The foundation lies in the Laplacian decomposition of time series:

$$L(t) = \sum_{k=-K}^{K} \mathbf{c}_k \cdot e^{2\pi i k t / T}, \tag{2}$$

where $L(t)$ represents the Laplacian motion field at time $t$, with $\mathbf{c}_k$ denoting coefficients. Through Euler's formula, we substitute this representation into our motion prediction:

$$L(t) = \sum_{k=-K}^{K} \mathbf{c}_k \cdot \cos\left(\frac{2\pi k t}{T}\right) + i \sum_{k=-K}^{K} \mathbf{c}_k \cdot \sin\left(\frac{2\pi k t}{T}\right). \tag{3}$$

This transformation naturally handles periodic motions common in dynamic scenes and the frequency components provide interpretable control over motion characteristics. To further enhance our motion representation, we extend the equation to a simple formulation:

$$L(t) = \sum_{k=0}^{K-1} \left[\alpha_k \cos(2\pi k t) + \beta_k \sin(2\pi k t)\right]. \tag{4}$$

Here, coefficients $(\alpha_k, \beta_k)$ are learnable parameters. This design aims to automatically adapt to scene-specific motion frequencies while maintaining end-to-end differentiability. The Laplacian decoding allows our model to learn appropriate frequency compositions directly from data space, eliminating the need for manual frequency band selection. The orthogonal basis properties enable stable gradient computation during optimization. Incorporation of learnable frequencies $f_k$ through gradient enhances the ability to capture different frequency motion components:

$$\frac{\partial \mathcal{L}}{\partial f_k} = \frac{1}{\sigma_k^2} \sum_t \left(\frac{\partial \mathcal{L}}{\partial L(t)} \cdot t \cdot [-\alpha_k \sin(2\pi f_k t) + \beta_k \cos(2\pi f_k t)]\right), \tag{5}$$

where $\sigma_k$ denotes temporal variance. This mechanism automatically balances frequency preservation with motion stability. Then we introduce an attention mechanism which combines Laplacian features with Hash spatial features $\mathcal{H}_s$:

$$\mathcal{A}_L(t) = L(t) \cdot \text{MLP}(\mathcal{H}_s). \tag{6}$$

### 3.1.3 MULTI-SCALE LAPLACIAN PYRAMID SUPERVISION

To enforce consistency across different frequency bands, we introduce a multi-scale supervision strategy that enforces consistency across frequency bands, enhancing the model's ability to better detail preservation. We supervise reconstruction using Laplacian pyramid decomposition:

$$\mathcal{L}_{\text{lap}} = \sum_{l=1}^{L} \lambda_l \|\mathcal{L}_l(I_{\text{render}}) - \mathcal{L}_l(I_{\text{gt}})\|_1, \tag{7}$$

where $\lambda_l$ decreases exponentially to emphasize finer details. This loss function encourages model to focus on both coarse and fine features, ensuring a comprehensive understanding of motion dynamics.

### 3.2 ENHANCED GAUSSIAN DYNAMICS ATTRIBUTE WITH ADAPTIVE REGULARIZATION

To effectively model the dynamic variations inherent in complex scenes, we augment the standard 3D Gaussian Splatting representation. Specifically, we associate each 3D Gaussian $G_i$ with a learnable dynamics attribute, denoted as $\mathbf{d}_i \in \mathbb{R}^{D_d}$, where $D_d$ represents the dimensionality of this attribute space. The dynamics attribute $\mathbf{d}_i$ is introduced to explicitly encapsulate these latent per-Gaussian conditional variations, providing a dedicated representation of dynamic properties.

To further improve the modeling of dynamic scene changes, we introduce a fusion mechanism that concatenates the original dynamic attribute vector $\mathbf{d}_i$ with the Hash temporal feature $\mathcal{H}_t$, forming an augmented feature representation $\tilde{\mathcal{H}}_t = \text{Concatenate}(\mathbf{d}_i, \mathcal{H}_t)$.

This concatenation provides a straightforward yet effective means of integrating scene-specific temporal information with the Gaussian's intrinsic dynamic attributes, enabling the model to leverage both sources for more accurate deformation prediction. To effectively combine spatial and temporal information, we introduce an attention mechanism to aggregate spatio-temporal features through:

$$\mathcal{A}_h(t) = \tilde{\mathcal{H}}_t \cdot \mathcal{H}_s. \tag{8}$$

### 3.2.1 ADAPTIVE DYNAMIC REGULARIZATION

To ensure that our method better model dynamic changes, we implement a selective regularization mechanism that targets only those Gaussians exhibiting "abnormally large" or "highly dynamic" changes. These gaussians are referred to as "outliers", which need to be increase their gradients and thus promote their deformation or dynamic transformations.

Instead of using fixed thresholds or applying a regularization on all gaussians, our method employs a data-driven, adaptive dynamic selection scheme. Specifically, for each Gaussian, we compute the Euclidean distance between its dynamic attribute $\mathbf{d}_i$ and a reference mean dynamic attribute $\bar{\mathbf{d}}$, as well as the associated standard deviation $d_i = \|\mathbf{d}_i - \bar{\mathbf{d}}\|_2$.

Let $\mu_{\text{dist}}$ and $\sigma_{\text{dist}}$ denote the mean and standard deviation of all $d_i$ across the Gaussian set. We then generate a mask to identify primitives that are significantly deviating from the normal scale:

$$\text{mask}_i = d_i > \mu_{\text{dist}} + \sigma_{\text{dist}}. \tag{9}$$

Only the Gaussians satisfying this outlier criterion—i.e., those with $\text{mask}_i = 1$—are subjected to the additional regularization loss:

$$\mathcal{L}_{\text{dy}} = \frac{1}{N} \sum_{i=1}^{N} \text{mask}_i \cdot d_i^2. \tag{10}$$

The purpose of this selective regularization is to intensify the gradients for Gaussians exhibiting large changes, thereby explicitly promoting their deformation and densification. By employing this dynamic regularization mechanism, the model adaptively concentrates regularization efforts on the most informative and dynamically relevant regions, effectively enhancing the capacity to model complex scene dynamics without imposing uniform constraints across all Gaussians.

In addition, we use Normalized cross-correlation (NCC) Yoo & Han (2009) loss $\mathcal{L}_{\text{NCC}}$ to evaluate the similarity between two images while maintaining invariance to brightness variations to enhance alignment accuracy. The total loss $\mathcal{L}$ used for training is made up of four distinct terms, each weighted by a corresponding hyperparameter $\lambda$ to control its contribution:

$$\mathcal{L} = \mathcal{L}_{\text{orig}} + \lambda_{\text{NCC}}\mathcal{L}_{\text{NCC}} + \lambda_{\text{lap}}\mathcal{L}_{\text{lap}} + \lambda_{\text{dy}}\mathcal{L}_{\text{dy}}, \tag{11}$$

where $\mathcal{L}_{\text{orig}}$ denotes original loss function of 3DGS and consists of $\mathcal{L}_1$ and Structural Similarity Index Measure (SSIM) loss functions Wang et al. (2004) $\mathcal{L}_{\text{SSIM}}$.

### 3.3 ADAPTIVE GAUSSIAN SPLIT STRATEGY

In this section, we address the challenge of optimizing Gaussian representations of motion dynamics. Our approach utilizes the analysis about the structure of each Gaussian to adaptively refine Gaussian parameters based on local neighborhood information, enhancing the model's ability to capture complex motion patterns.

#### 3.3.1 KDTREE-BASED PRIMITIVE ANALYSIS

To maintain spatial coherence and prevent overfitting, we analyze Gaussian primitives through their neighborhood relationships based on Euclidean distance. By examining the size and anisotropy of each Gaussian Xie et al. (2024), we can determine which Gaussians exhibit significant differences in their motion characteristics. Covariance differences are computed through L2 norm:

$$\Delta\Sigma_{ij} = \|\Sigma_i - \Sigma_j\|_2. \tag{12}$$

This adaptive approach ensures that our models remain responsive to local variations in motion dynamics. By focusing on Gaussians with notable differences in size and anisotropy, we can selectively choose which Gaussian to split, thereby enhancing the model's ability to represent complex motion patterns without introducing unnecessary complexity.

#### 3.3.2 KL-DIVERGENCE GUIDED ADAPTATION

In some cases, we observed that the KDTree-based partitioning method can not accurately identify the dynamic Gaussian, which leads to the instability to capture dynamic motions. This is because the strategy of splitting Gaussian based on hard threshold will stop deriving when the shape and size of Gaussian in the neighborhood are similar. To further refine the Gaussian split process, we compute the KL-divergence between the neighbor Gaussian distribution $P$ and a uniform distribution $Q$:

$$D_{\text{KL}}(P \parallel Q) = \sum_{k=1}^{K} P(k) \log \frac{P(k)}{Q(k)} \tag{13}$$

The adaptive splitting threshold $\tau = \Delta\Sigma + D_{\text{KL}} \cdot \tau_{\text{base}}$, where $\tau_{\text{base}}$ is a hyperparameter. This mechanism allows for dynamic adjustments to the model complexity based on the observed motion patterns. Through adaptive dynamic Gaussian optimization strategy, we can determine when a Gaussian should be split more effectively, ensuring the model captures the nuances of motion dynamics while maintaining computational efficiency. This strategy enhances the robustness against overfitting by focusing on the most relevant Gaussian structures.

Table 1: **Quantitative comparison to previous methods on HyperNeRF Park et al. (2021b) dataset.** The higher PSNR(↑) and higher SSIM(↑) denote better rendering quality. The color of each cell shows the best and the second best.

| Scene Method | broom2 SSIM↑ | PSNR↑ | LPIPS↓ | vrig-3dprinter SSIM↑ | PSNR↑ | LPIPS↓ | vrig-chicken SSIM↑ | PSNR↑ | LPIPS↓ | vrig-peel-banana SSIM↑ | PSNR↑ | LPIPS↓ |
|---|---|---|---|---|---|---|---|---|---|---|---|---|
| HyperNeRF Park et al. (2021b) | 0.210 | 19.51 | — | 0.635 | 20.04 | — | 0.828 | 27.46 | — | 0.719 | 22.15 | — |
| D3DGS Yang et al. (2024) | 0.269 | 19.99 | 0.700 | 0.656 | 20.71 | 0.277 | 0.640 | 22.77 | 0.363 | 0.853 | 25.95 | 0.155 |
| MotionGS Zhu et al. (2024) | 0.380 | 22.30 | — | 0.710 | 21.80 | — | 0.790 | 26.80 | — | 0.690 | 28.20 | — |
| MoDec-GS Kwak et al. (2025) | 0.303 | 21.04 | 0.666 | 0.706 | 22.00 | 0.265 | 0.834 | 28.77 | 0.197 | 0.873 | 28.25 | 0.173 |
| 4DGaussians Wu et al. (2024) | 0.366 | 22.01 | 0.557 | 0.705 | 21.98 | 0.327 | 0.806 | 28.49 | 0.297 | 0.847 | 27.73 | 0.204 |
| ED3DGS Bae et al. (2024) | 0.371 | 21.84 | 0.531 | 0.715 | 22.34 | 0.294 | 0.836 | 28.75 | 0.185 | 0.867 | 28.80 | 0.178 |
| Grid4D Xu et al. (2024) | 0.414 | 21.78 | 0.423 | 0.723 | 22.33 | 0.245 | 0.848 | 29.27 | 0.199 | 0.875 | 28.44 | 0.167 |
| Ours | 0.422 | 22.36 | 0.413 | 0.724 | 22.56 | 0.264 | 0.858 | 29.57 | 0.166 | 0.876 | 28.81 | 0.169 |

Table 2: **Quantitative comparison to previous methods on D-NeRF Pumarola et al. (2021) dataset.** The color of each cell shows the best and the second best. More detail results can be found in supplementary material.

| Method | SSIM↑ | PSNR↑ | LPIPS↓ |
|---|---|---|---|
| 3DGS Kerbl et al. (2023) | 0.930 | 23.40 | 0.077 |
| K-Planes Fridovich-Keil et al. (2023) | 0.970 | 31.41 | 0.047 |
| HexPlane Cao & Johnson (2023) | 0.972 | 31.92 | 0.038 |
| 4DGaussians Wu et al. (2024) | 0.985 | 35.32 | 0.021 |
| D3DGS Yang et al. (2024) | 0.991 | 40.08 | 0.013 |
| SC-GS Huang et al. (2024) | 0.993 | 41.66 | 0.009 |
| Grid4D Xu et al. (2024) | 0.994 | 41.99 | 0.008 |
| Ours | 0.994 | 42.17 | 0.007 |

# 4 EXPERIMENT

## 4.1 EXPERIMENT SETUP

We evaluate our method using three widely recognized datasets, comprising two real-world datasets and one synthetic dataset. The Neu3D Li et al. (2022) dataset is a real-world collection that features multiple static cameras and includes between 18 to 21 multi-view videos. We generate 300 frames for each video and initial point clouds for each scene following 4DGaussians Wu et al. (2024). HyperNeRF Park et al. (2021b) is a real-world dataset that captures continuous views with intricate topological variations at each timestamp within a dynamic scene. In our experiment, we utilized the "vrig" subset, which was captured using stereo cameras, training the model with data from one camera while validating it with data from the other. The D-NeRF Pumarola et al. (2021) dataset serves as a synthetic dataset tailored for monocular scenes, with each scene comprising between 50 to 200 frames. Due to discrepancies between the training and testing scenarios in the Lego subset of the D-NeRF Pumarola et al. (2021) dataset, we excluded it from our experimental analysis.

## 4.2 COMPARISONS

On the Neu3D dataset, our approach demonstrates exceptional proficiency as shown in Tab. 3. The primary challenge here lies in accurately modeling intricate, often non-rigid, temporal dynamics while simultaneously reconstructing high-fidelity static scene geometry from these fixed perspectives. Our method excels in generating temporally coherent motion representations and preserving sharp geometric details throughout the sequences, effectively disentangling dynamic elements from the static background. In contrast, competing methods frequently struggle to maintain long-term temporal consistency across the multiple views, often exhibiting noticeable motion blur, particularly during complex actions or over extended durations.

The HyperNeRF "vrig" subset introduces a distinct set of demanding conditions. This dataset tests the model's ability to handle complex motion while maintaining consistency across stereo viewpoints and adapt to evolving scene topology. Our method showcases remarkable resilience and adaptability in handling these extreme deformations and effectively leverages the stereo information, generalizing robustly across the viewpoints even when trained on one and validated on the

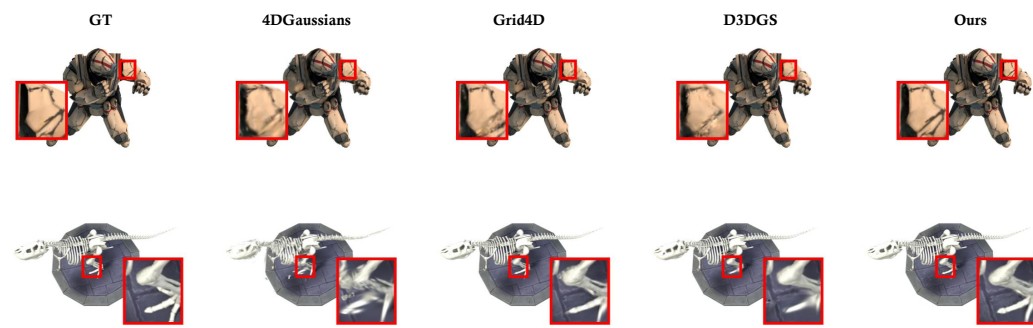

Figure 3: **Comparison Results.** Visual differences are highlighted with red insets for better clarity. Our approach consistently outperforms on D-NeRF Pumarola et al. (2021) dataset, demonstrating clear advantages in thin geometries and fine-scale details scenarios. Best viewed in color.

Table 3: **Quantitative comparison to previous methods on Neu3D Li et al. (2022) dataset.** Color of each cell shows the best and the second best. We show the average results of all scenes. More detail results can be found in supplementary material.

| Method | SSIM↑ | PSNR↑ | LPIPS↓ |
|---|---|---|---|
| 4DGaussians Wu et al. (2024) | 0.935 | 30.36 | 0.152 |
| Grid4D Xu et al. (2024) | 0.934 | 30.50 | 0.147 |
| Spacetime Li et al. (2024) | 0.944 | 31.46 | 0.142 |
| ED3DGS Bae et al. (2024) | 0.943 | 31.92 | 0.139 |
| Ours | 0.944 | 32.12 | 0.134 |

other as shown in Tab. 1 and Fig. 5. It consistently reconstructs intricate topological changes with greater accuracy and fewer visual artifacts or geometric distortions compared to existing approaches.

Furthermore, evaluation on the synthetic D-NeRF dataset underscores our method's inherent strength in inferring coherent 3D structure and plausible motion from limited input as shown in Tab. 2 and Fig. 3. Reconstructing dynamic 3D geometry from a single, potentially moving, camera viewpoint over time presents profound depth ambiguities and necessitates strong priors and temporal reasoning. Despite this inherent ill-posedness and the scarcity of explicit geometric cues, our approach generates remarkably temporally stable and geometrically plausible reconstructions. Consequently, it significantly outperforms baseline methods, which, under these monocular constraints, often exhibit pronounced depth inaccuracies that betray instabilities in their representation.

Across this diverse range of evaluated datasets, our method consistently achieves a marked superiority in performance. This advantage is evident in both the final reconstruction fidelity and the accurate, coherent capture of dynamic motion, ranging from subtle deformations to large-scale topological changes. This consistent success across varied and demanding conditions robustly validates the effectiveness, versatility, and broad applicability of our method for 4D reconstruction.

### 4.3 ABLATION STUDY AND ANALYSIS

To validate the effectiveness of each component within our framework, we conduct comprehensive ablation studies to validate the necessity of each component as shown in Tab. 4 and Fig. 4.

#### 4.3.1 EFFECT OF LAPLACIAN MOTION FLOW EXTRACTION MODULE

Replacing this module with a defrom MLP leads to poorer performance, especially in scenes with diverse motion patterns or objects of varying sizes evolving over time. Dynamic scenes inherently possess variations across multiple spatial and temporal scales. This module is designed to capture these hierarchies effectively. It allows the model to represent fine details of motion trajectories while enabling the modeling of slow, gradual changes. By processing information hierarchically, it ensures consistent and accurate representation of scene dynamics across different frequency.

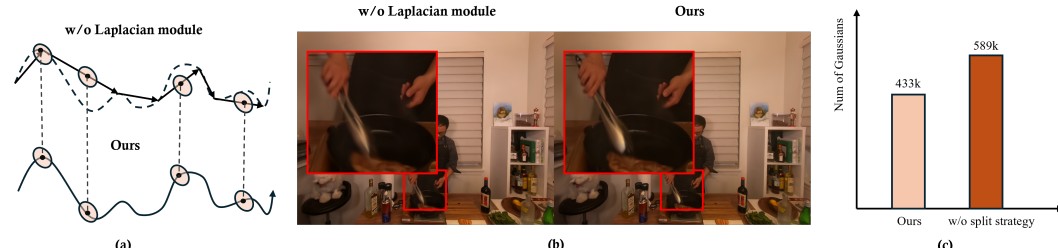

Figure 4: **Ablation Results.** Replacing Laplacian motion flow extraction module leads to poorer performance. Besides, the split strategy helps in reducing the number of final Gaussians.

Table 4: **Ablation evaluation on Neu3D Li et al. (2022) dataset.**

| Method | SSIM↑ | PSNR↑ | LPIPS↓ |
|---|---|---|---|
| w/o Laplacian module | 0.938 | 31.64 | 0.149 |
| w/o dynamic attribute | 0.938 | 31.72 | 0.147 |
| w/o adaptive split strategy | 0.943 | 31.96 | **0.133** |
| w/o $\mathcal{L}_{lap}$ | 0.939 | 31.70 | 0.148 |
| Ours | **0.944** | **32.12** | 0.134 |

### 4.3.2 EFFECT OF ADAPTIVE GAUSSIAN SPLIT STRATEGY AND LAPLACIAN PYRAMID LOSS

Removing this component and reverting to original 3DGS split strategy results in a drop in reconstruction quality. This component improves render quality by intelligently allocating Gaussian primitives through densifying regions with high dynamics while pruning redundant primitives. This leads to a more compact representation of the scene and set of Gaussians compared to non-adaptive methods while achieving similar quality as shown in Tab. 4.

When this loss is removed on the rendered image, we observe a noticeable degradation in reconstruction quality. The Laplacian pyramid loss decomposes the reconstruction error across multiple frequency bands by comparing the Laplacian pyramids of the rendered and ground truth images. This loss function proves essential because it enforces structural consistency across different scales, effectively preserving fine details that would otherwise be lost in single-scale supervision.

### 4.3.3 EFFECT OF GAUSSIAN DYNAMICS ATTRIBUTE

Compared with full model, removing the Gaussian dynamics attribute leads to poorer performance. This difference underscores the importance of embedding dedicated dynamic attributes within the Gaussians themselves. By incorporating this, we allow each Gaussian to better adapt its shape and orientation according to its specific local dynamics, effectively capturing details and mitigating the feature collision issues inherent in relying solely on lower-rank grids.

## 5 CONCLUSION

In this paper, we present a novel approach for dynamic 3DGS that addresses the challenges of anisotropic spatio-temporal sampling through a hybrid explicit-implicit encoding framework. We introduced three key innovations: Firstly, a hybrid Laplacian motion flow extraction module combining Laplacian analysis with Hash encoding, effectively decoupling different motion frequencies from complex deformation details. Secondly, an enhanced Gaussian dynamics attribute that compensates for highly dynamic areas induced by geometric deformation. Thirdly, an adaptive Gaussian split strategy guided by KDTree-based analysis, which automatically adjusts dynamic primitive density and anisotropy.This work advance the state-of-the-art in dynamic scene modeling by bridging the gap between explicit representations and spectral analysis, with potential applications in VR/AR and scene reconstruction.

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

## A    ETHICS STATEMENT

This work adheres to the ICLR Code of Ethics. In this study, no human subjects or animal experimentation was involved. All datasets used were sourced in compliance with relevant usage guidelines, ensuring no violation of privacy. We have taken care to avoid any biases or discriminatory outcomes in our research process. No personally identifiable information was used, and no experiments were conducted that could raise privacy or security concerns. We are committed to maintaining transparency and integrity throughout the research process.

## B    REPRODUCIBILITY STATEMENT

We have made every effort to ensure that the results presented in this paper are reproducible. All code and datasets will be made publicly available after the paper is accepted to facilitate replication and verification. The experimental setup, including training steps, model configurations, and hardware details, is described in detail in the paper. We believe these measures will enable other researchers to reproduce our work and further advance the field.

## C    LLM USAGE STATEMENT

Large Language Models (LLMs) were used to aid in the writing and polishing of the manuscript. Specifically, we used an LLM to assist in refining the language, improving readability, and ensuring clarity in various sections of the paper. The model helped with tasks such as sentence rephrasing, grammar checking, and enhancing the overall flow of the text.

It is important to note that the LLM was not involved in the ideation, research methodology, or experimental design. All research concepts, ideas, and analyses were developed and conducted by the authors. The contributions of the LLM were solely focused on improving the linguistic quality of the paper, with no involvement in the scientific content or data analysis.

We have ensured that the LLM-generated text adheres to ethical guidelines and does not contribute to plagiarism or scientific misconduct.

## D    EXPERIMENTS SETTING DETAILS

Our framework is implemented on PyTorch and all experiments are conducted on a single RTX A6000 GPU. The scheduler of the learning rate primarily follows D3DGS Yang et al. (2024) and Grid4D Xu et al. (2024). Large Language Model (LLM) was employed to polish and refine the grammatical structure of this paper.

For training, we set the number of iterations to 30k for Neu3D Li et al. (2022), 30-40k for HyperNeRF Park et al. (2021b), 50k for D-NeRF Pumarola et al. (2021). The dimension resolution of the hash grids, the number of Laplacian coefficients, and the dimension of neighboring Gaussians used in the split strategy are adapted according to the scale of each scene. The Deformed MLP consists of a single-layer MLP with a width of 256, followed by a ReLU activation function. We implement 20 neighbors for dynamic regularization and configure the dynamic attribute with a dimension varying from 16 to 64 based on scenes, using a regularization coefficient of $\lambda_{\text{dy}} = 0.1$. The Laplacian-based loss is introduced starting from the 3k iteration, while the Normalized Cross-Correlation (NCC) loss Yoo & Han (2009) is applied from the 20k iteration onward. The loss weights are set to $\lambda_{\text{lap}} = 1$ and $\lambda_{\text{NCC}} = 0.01$ for common scene.

For D-NeRF Pumarola et al. (2021) dataset, the background is set to black, consistent with the settings in D3DGS Yang et al. (2024) and Grid4D Xu et al. (2024). For the scenes in the Neu3D Li et al. (2022), HyperNeRF Park et al. (2021b) dataset, we utilize the SFM Schonberger & Frahm (2016) points created from COLMAP Schönberger et al. (2016) to initialize Gaussians.

# E  THEORETICAL DERIVATION FOR LAPLACIAN MOTION FLOW EXTRACTION

We denote the dynamic motion field (per-point displacement) as a function over the 3D spatial domain and time:

$$\mathbf{u}(\mathbf{x}, t) \in \mathbb{R}^3, \tag{14}$$

Stack per-Gaussian signals into a matrix/time series $U(t) = [\mathbf{u}_1(t), \ldots, \mathbf{u}_N(t)]^\top \in \mathbb{R}^{N \times 3}$. For clarity, we treat one scalar channel and suppress the color and axis dimension.

## E.1  LOW-RANK DECOMPOSITION

Assume we attempt to model the spatio-temporal field by a separable low-rank decomposition:

$$U(t) = PS(t) \approx \sum_{r=1}^{R} \mathbf{p}_r s_r(t), \tag{15}$$

where $P \in \mathbb{R}^{N \times R}$ and $S(t) \in \mathbb{R}^R$. This is equivalent to approximating each instantaneous spatial snapshot by an R-rank matrix. Any truncation to rank $R$ eliminates singular vector directions with small singular values through SVD. Those small singular values often correspond to high spatial-frequency or localized deviations.

Let the per-time snapshot have spectral decomposition $U(t) = \sum_{k=1}^{N} \sigma_k(t) \mathbf{u}_k(t) \mathbf{v}_k^\top(t)$. Truncating to $R$ keeps only $\sigma_1 \ldots \sigma_R$. Because the energy of smooth, large-scale deformations concentrates in leading singular values, truncation discards small-energy but high-frequency components. Therefore, if approximating dynamics by low-rank temporal bases, the model cannot reconstruct rapid localized non-rigid motion that manifests as high-frequency components in either the spatial graph or temporal spectrum.

## E.2  GRID SAMPLING AND HASH ENCODINGS

Consider encoding features on a discrete grid or hashed table over $(x, y, z, t)$. The discrete sampling operator $S$ maps a continuous field $u(\mathbf{x}, t)$ to sampled values $u_s = S[u]$. According to Nyquist sampling theory, if the continuous field has frequency content above the sampling Nyquist limit, aliasing occurs and multiple different frequency components map to the same discrete samples.

The finite capacity of hash tables can cause different spatially close signals to map to similar indices, which acts like a low-pass operator in practice or unpredictable mixing of features. If the spacing of the sampling grid $h$ is such that the maximum representable spatial frequency is $\omega_{\max} \approx \pi/h$, any true signal component with $|\omega| > \omega_{\max}$ will be observed as an aliased lower frequency $\omega_a$ satisfying $\omega_a = \omega - m \cdot (2\pi/h)$ for some integer $m$. That collapsed mapping between distinct frequencies is a collision in spectral domain caused by non-orthogonality between bases. High-dimensional encoding without spectral priors can produce inconsistent aliasing where different time frames map features to the same code.

Therefore, the low-rank or strong smoothing regularization enforces deformation consistency but attenuates the high frequencies. Conversely, high-capacity hash encodings can represent high frequencies, but without spectral regularization they produce inconsistent aliasing and feature collisions. We want to preserve high-frequency local motion details while maintaining smooth temporal coherence. We achieve this by decomposing the per-Gaussian signal into Laplacian eigenmodes.

## E.3  LAPLACIAN EIGENBASIS

For Laplacian operator for graph structure or continuous field, we have:

$$\Delta f(x) = \nabla^2 f(x). \tag{16}$$

Eigenvectors $u_k$ are orthonormal spatial modes ordered by increasing spatial frequency. Therefore, we can decompose the motion field into:

$$U(x,t) = \sum_k \alpha_k(t)\phi_k(x),$$ (17)

where $\alpha_k$ is obtained by Laplacian expansion $\alpha_k(t) = \sum_{m=-M}^{M} c_{k,m} e^{i2\pi mt/T}$ and the spatial basis $\phi_k(x)$ is orthogonal in $L^2$ space.

Laplacian gives an explicit spectral weighting, enabling us to optimize crucial frequency bands. Instead of hard low-rank truncation, spectral methods can preserve high-frequency coefficients at current time windows and penalize incoherent high frequencies across neighbors by shaping the penalty rather than an all-or-nothing rank truncation.

We use Laplacian spectrum expansion to model per-Gaussian dynamics. In this way, each dynamic attribute is decoupled into orthogonal components of different frequencies to ensure that it can express high-frequency details without interfering with each other in optimization.

## F ADAPTIVE GAUSSIAN SPLIT STRATEGY DETAILS

We apply KL-divergence in split strategy by computing the KL-divergence between the neighbor Gaussian distribution $P$ and a uniform distribution $Q$:

$$D_{\text{KL}}(P \parallel Q) = \sum_{k=1}^{K} P(k) \log \frac{P(k)}{Q(k)}.$$ (18)

While the original notation used $P$ and $Q$ to suggest probability distributions, in our implementation they are instantiated as *zero-mean multivariate Gaussian distributions* with covariances $\Sigma_i$ (for the candidate Gaussian) and $\bar{\Sigma}$ (mean of neighboring Gaussians), respectively. Specifically:

$$D_{KL}(N(0,\Sigma_i)\|N(0,\bar{\Sigma})) = \frac{1}{2}\left[\text{tr}(\bar{\Sigma}^{-1}\Sigma_i) - k + \log \frac{\det \bar{\Sigma}}{\det \Sigma_i}\right].$$ (19)

To enhance stability and avoid directional bias, we adopt the *symmetric KL divergence*:

$$D_{\text{KL}} = D_{KL}(\Sigma_i\|\bar{\Sigma}) + D_{KL}(\bar{\Sigma}\|\Sigma_i).$$ (20)

The adaptive splitting threshold $\tau$ becomes:

$$\tau = \Delta\Sigma + D_{\text{KL}} \cdot \tau_{\text{base}}.$$ (21)

We set hyperparameter $\tau_{\text{base}} = 1e-4$. It serves as a scene-dependent parameter to determine when a candidate Gaussian significantly deviates from its local neighbors. A lower value allows more aggressive splitting in scenes with fine or deformable structures. To validate its effectiveness, we include an ablation in Tab. 5 to comparing results with different $\tau_{\text{base}}$.

Table 5: **The comparison results of different hyperparameter $\tau_{\text{base}}$ on D-NeRF datasets.**

| $\tau_{\text{base}}$ | 0 | | | 1e-2 | | | 1e-4 | | |
|---|---|---|---|---|---|---|---|---|---|
| | SSIM | PSNR | LPIPS | SSIM | PSNR | LPIPS | SSIM | PSNR | LPIPS |
| Ours | 0.994 | 42.07 | 0.008 | 0.994 | 42.01 | 0.008 | 0.994 | 42.17 | 0.007 |

## G MORE EXPERIMENT RESULTS

In this section, more detail experiment results will be reported. Tab. 7, Tab. 8, Tab. 9 and Tab. 10 show the PSNR, SSIM and LPIPS results on dataset. Tab. 7 and Tab. 8 presents the detail results

on Neu3D Li et al. (2022) dataset. Tab. 9 and Tab. 10 show the experiment results on the synthetic scenes of D-NeRF Pumarola et al. (2021) dataset. In addition, we add more rendering in Fig. 6 for HyperNeRF Park et al. (2021b) and Fig. 2 for Neu3D Li et al. (2022) .

Our method demonstrates consistent superiority across all benchmarks, excelling where others struggle. On Neu3D (Tab. 7 and Fig. 2), our method adeptly handles complex temporal dynamics and reconstructs high-fidelity static geometry. It delivers temporally coherent motion, sharp details, and effective separation of dynamic and static elements, contrasting with competing approaches that often show motion blur, ghosting artifacts, or geometric distortions.

The HyperNeRF "vrig" subset (Fig. 6) show our strong adaptability to leverage stereo information for consistent reconstructions and generalization across views, with fewer visual artifacts and distortions than alternatives. For synthetic scene reconstruction on D-NeRF (Tab. 9 and Tab. 10), our method excels in inferring coherent 3D structures and motion from limited monocular inputs, overcoming depth ambiguities through robust temporal reasoning. This cross-dataset validation demonstrates that our framework uniquely combines geometric accuracy, temporal stability, and view consistency, validating its effectiveness and establishing state-of-the-art performance in dynamic scene reconstruction.

In addition, we provide detailed comparison with Grid4D Xu et al. (2024) on the D-NeRF dataset in Tab. 6, including model size, training time, and inference speed. As we can see, on the D-NeRF dataset, our model has 51MB, slightly larger than Grid4D (50MB). This size increase is due to our incorporation of Laplacian-based frequency modeling and dynamic appearance embedding, which offer improved flexibility and fidelity for dynamic scenes.

Moreover, our method requires about 2 hours per scene, which is a little longer than Grid4D (1h). Furthermore, our approach achieves 169 FPS at 800×800 resolution on D-NeRF scenes, enabling real-time rendering. While Grid4D reaches 180 FPS, it does not model dynamic attributes or frequency-aware motion. In contrast, our method achieves higher PSNR and better visual quality, offering a better quality-efficiency trade-off.

Table 6: **Detailed comparison on D-NeRF datasets about model size, training time and the rendering FPS of our method.**

|        | SSIM  | PSNR  | PSNR  | Szie | Time   | FPS |
|--------|-------|-------|-------|------|--------|-----|
| Grid4D | 0.994 | 41.99 | 0.008 | 50MB | 1h5min | 180 |
| Ours   | 0.994 | 42.17 | 0.007 | 51MB | 2h6min | 169 |

Table 7: **Quantitative comparison to previous methods on Neu3D Li et al. (2022) dataset.** The higher PSNR(↑) and higher SSIM(↑) denote better rendering quality. The color of each cell shows the  best .

| Scene | coffee_martini | | | cook_spinach | | | cut_roasted_beef | | |
|-------|-------|-------|--------|-------|-------|--------|-------|-------|--------|
|       | SSIM↑ | PSNR↑ | LPIPS↓ | SSIM↑ | PSNR↑ | LPIPS↓ | SSIM↑ | PSNR↑ | LPIPS↓ |
| 4DGaussians Wu et al. (2024) | 0.910 | 28.52 | 0.162 | 0.946 | 32.46 | 0.153 | 0.938 | 31.01 | 0.153 |
| Grid4D Xu et al. (2024) | 0.896 | 27.77 | 0.177 | 0.948 | 32.03 | 0.144 | 0.943 | 31.41 | 0.142 |
| Spacetime Li et al. (2024) | 0.914 | 28.55 | 0.158 | 0.954 | 32.39 | 0.141 | 0.952 | 32.70 | 0.143 |
| ED3DGS Bae et al. (2024) | 0.917 | 29.26 | 0.147 | 0.948 | 32.61 | 0.146 | 0.951 | 33.64 | 0.144 |
| Ours | 0.918 | 29.60 | 0.147 | 0.952 | 32.74 | 0.133 | 0.954 | 33.37 | 0.132 |

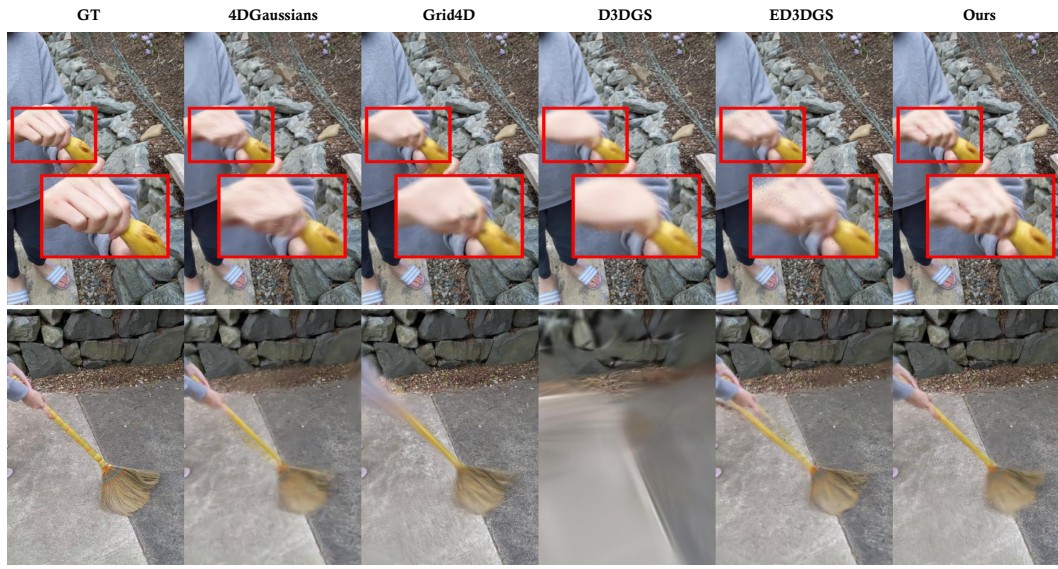

Figure 5: **Comparison Results.** Our approach consistently outperforms on HyperNeRF Park et al. (2021b) dataset, demonstrating advantages in challenging scenarios. Best viewed in color.

Table 8: **Quantitative comparison to previous methods on Neu3D Li et al. (2022) dataset.** The higher PSNR(↑) and higher SSIM(↑) denote better rendering quality. The color of each cell shows the best .

| Scene | flame_salmon_1 | | | flame_steak | | | sear_steak | | |
|---|---|---|---|---|---|---|---|---|---|
| | SSIM↑ | PSNR↑ | LPIPS↓ | SSIM↑ | PSNR↑ | LPIPS↓ | SSIM↑ | PSNR↑ | LPIPS↓ |
| 4DGaussians Wu et al. (2024) | 0.913 | 29.33 | 0.159 | 0.946 | 29.40 | 0.146 | 0.954 | 31.42 | 0.136 |
| Grid4D Xu et al. (2024) | 0.911 | 29.24 | 0.159 | 0.952 | 30.10 | 0.132 | 0.957 | 32.47 | 0.131 |
| Spacetime Li et al. (2024) | 0.921 | 28.59 | 0.147 | 0.961 | 32.79 | 0.132 | 0.964 | 33.76 | 0.130 |
| ED3DGS Bae et al. (2024) | 0.926 | 29.96 | 0.134 | 0.957 | 32.53 | 0.128 | 0.958 | 33.47 | 0.131 |
| Ours | 0.921 | 30.00 | 0.137 | 0.958 | 33.12 | 0.124 | 0.958 | 33.89 | 0.132 |

Table 9: **Quantitative comparison to previous methods on D-NeRF Pumarola et al. (2021) dataset.** The higher PSNR(↑) and higher SSIM(↑) denote better rendering quality. The color of each cell shows the best and the second best .

| Scene | bouncingballs | | | hellwarrior | | | hook | | | jumpingjacks | | |
|---|---|---|---|---|---|---|---|---|---|---|---|---|
| | SSIM↑ | PSNR↑ | LPIPS↓ | SSIM↑ | PSNR↑ | LPIPS↓ | SSIM↑ | PSNR↑ | LPIPS↓ | SSIM↑ | PSNR↑ | LPIPS↓ |
| 3DGS Kerbl et al. (2023) | 0.959 | 23.20 | 0.060 | 0.916 | 29.89 | 0.106 | 0.888 | 21.71 | 0.103 | 0.930 | 20.64 | 0.083 |
| K-Planes Fridovich-Keil et al. (2023) | 0.993 | 40.05 | 0.032 | 0.952 | 24.58 | 0.082 | 0.949 | 28.12 | 0.066 | 0.971 | 31.11 | 0.047 |
| HexPlane Cao & Johnson (2023) | 0.992 | 40.36 | 0.031 | 0.944 | 24.30 | 0.073 | 0.955 | 28.26 | 0.052 | 0.974 | 31.74 | 0.036 |
| 4DGaussians Wu et al. (2024) | 0.994 | 40.78 | 0.014 | 0.974 | 28.86 | 0.037 | 0.976 | 32.82 | 0.027 | 0.986 | 35.41 | 0.020 |
| D3DGS Yang et al. (2024) | 0.996 | 41.36 | 0.009 | 0.987 | 41.34 | 0.025 | 0.985 | 36.86 | 0.017 | 0.989 | 37.43 | 0.013 |
| SC-GS Huang et al. (2024) | 0.995 | 41.59 | 0.009 | 0.989 | 42.19 | 0.019 | 0.990 | 38.79 | 0.011 | 0.992 | 39.34 | 0.008 |
| Grid4D Xu et al. (2024) | 0.996 | 42.62 | 0.008 | 0.991 | 42.85 | 0.015 | 0.990 | 38.89 | 0.009 | 0.993 | 39.37 | 0.008 |
| Ours | 0.996 | 42.72 | 0.008 | 0.991 | 43.02 | 0.014 | 0.991 | 39.20 | 0.009 | 0.993 | 39.45 | 0.008 |

Table 10: **Quantitative comparison to previous methods on D-NeRF Pumarola et al. (2021) dataset.** The higher PSNR(↑) and higher SSIM(↑) denote better rendering quality. The color of each cell shows the best and the second best .

| Scene | mutant | | | standup | | | trex | | | Average | | |
|---|---|---|---|---|---|---|---|---|---|---|---|---|
| | SSIM↑ | PSNR↑ | LPIPS↓ | SSIM↑ | PSNR↑ | LPIPS↓ | SSIM↑ | PSNR↑ | LPIPS↓ | SSIM↑ | PSNR↑ | LPIPS↓ |
| 3DGS Kerbl et al. (2023) | 0.934 | 24.53 | 0.058 | 0.930 | 21.91 | 0.079 | 0.954 | 21.93 | 0.049 | 0.930 | 23.40 | 0.077 |
| K-Planes Fridovich-Keil et al. (2023) | 0.971 | 32.50 | 0.036 | 0.979 | 33.10 | 0.031 | 0.974 | 30.43 | 0.034 | 0.970 | 31.41 | 0.047 |
| HexPlane Cao & Johnson (2023) | 0.982 | 33.66 | 0.028 | 0.983 | 34.12 | 0.019 | 0.976 | 31.01 | 0.028 | 0.972 | 31.92 | 0.038 |
| 4DGaussians Wu et al. (2024) | 0.988 | 37.68 | 0.016 | 0.990 | 37.97 | 0.014 | 0.984 | 33.75 | 0.022 | 0.985 | 35.32 | 0.021 |
| D3DGS Yang et al. (2024) | 0.994 | 42.09 | 0.007 | 0.994 | 43.79 | 0.008 | 0.993 | 37.67 | 0.010 | 0.991 | 40.08 | 0.013 |
| SC-GS Huang et al. (2024) | 0.996 | 43.43 | 0.005 | 0.997 | 46.72 | 0.004 | 0.994 | 39.53 | 0.009 | 0.993 | 41.66 | 0.009 |
| Grid4D Xu et al. (2024) | 0.996 | 43.94 | 0.004 | 0.997 | 46.28 | 0.004 | 0.995 | 40.01 | 0.008 | 0.994 | 41.99 | 0.008 |
| Ours | 0.997 | 44.13 | 0.003 | 0.997 | 46.59 | 0.003 | 0.995 | 40.08 | 0.007 | 0.994 | 42.17 | 0.007 |

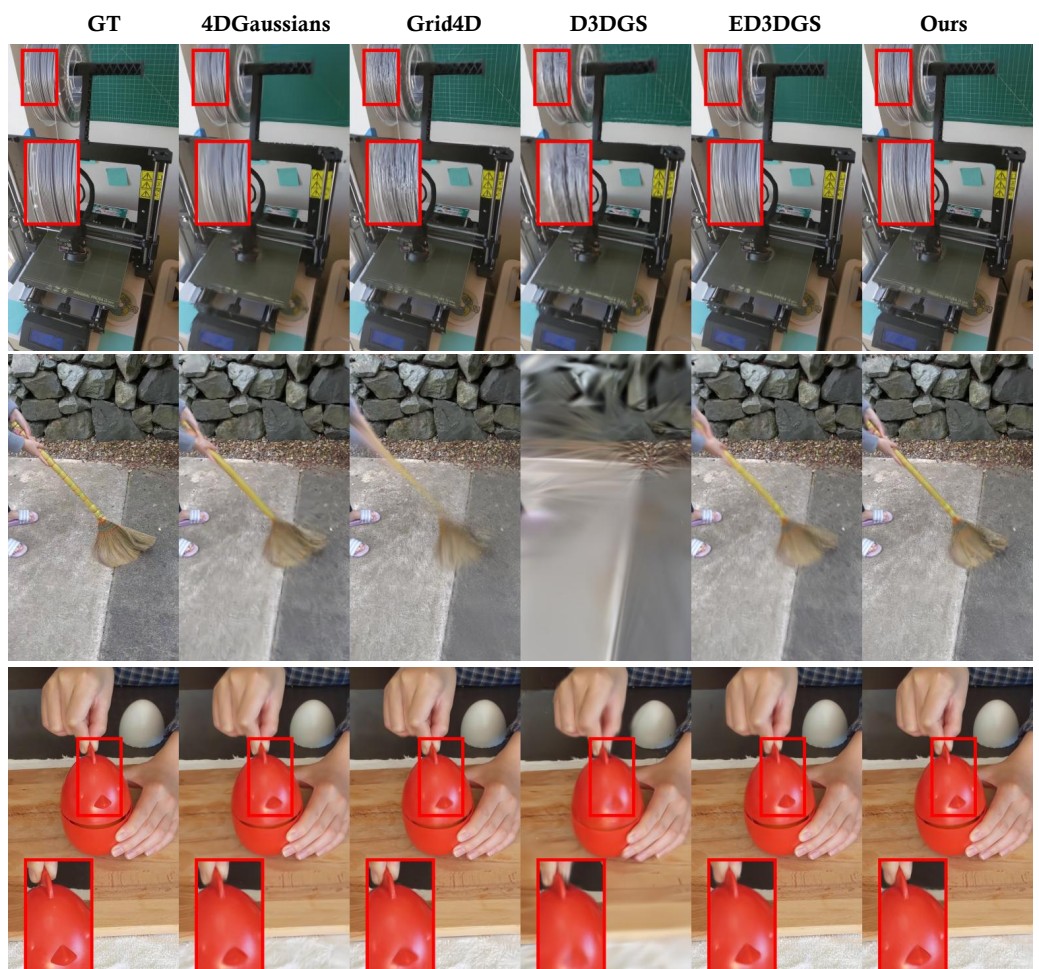

Figure 6: **More Comparison Results on HyperNeRF Park et al. (2021b) dataset.** Best viewed in color.

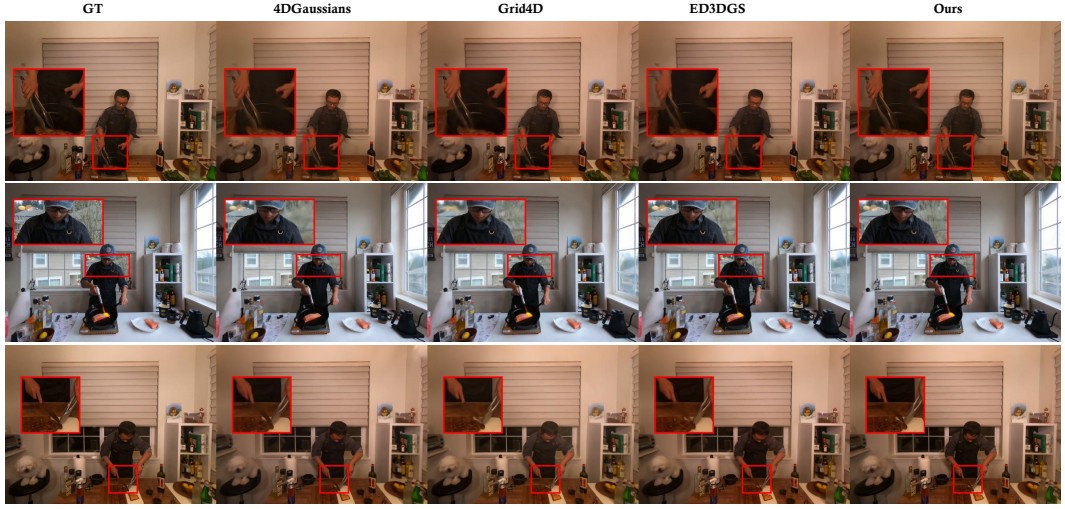

Figure 7: **More Comparison Results on Neu3D Li et al. (2022) dataset.** Best viewed in color.

