# OpenReview forum: "Laplacian Analysis Meets Dynamics Modelling: Gaussian Splatting for 4D Scene Reconstruction"
_ICLR.cc/2026/Conference — ICLR 2026 Conference Withdrawn Submission_

### Official Review · Reviewer_6NAJ · 2025-10-20

**Soundness:** 2
**Presentation:** 3
**Contribution:** 2
**Rating:** 4
**Confidence:** 4

**Summary:**

This paper proposes a new paradigm for 4D dynamic scene reconstruction. This paradigm introduces three new components: a spectral Laplacian motion flow representation which replaces the previous direct motion representation, an enhanced per-Gaussian dynamics attribute with adaptive regularization, and an adaptive Gaussian split derived from KDTree and KL divergence. These components jointly enhance the representation capability of 4D modeling and provides better performances than previous methods.

**Strengths:**

1. The motion Laplacian decomposition serves as a novel motion representation, where the learnable parameters are transformed from rigid movements to high-complexity Laplacian parameters. It is assumed that this representation has the capability to hold more vivid and flexible motions.
2. The proposed method achieves higher visual performances compared to previous baselines, illustrating the effectiveness of the proposed method.

**Weaknesses:**

1. Motivation and novelty. Although the Laplacian decomposition serves as a novel replacement over previous direct motion modeling method, the other contributions are neither well motivated nor correlated to main novelty. The per-Gaussian dynamics embedding is an adaptive variant of previous work [R1]. The KL divergence in 4D optimization is previously used in [R2].
2. Unclear attention mechanism. It is unclear how the 'attention mechanism' works in this paradigm. Is this a multiplication between two features? How it impacts the modeling capability?
3. The proposed method enhances the representation capability with more complex motion representation while the convergence speed is compromised, leading to unneglectable sacrifice.
4. Baselines. Recent advancements on 4D representation based on Gaussian Splatting are not considered [R3]. Previous advanced method 4DGS is not included into consideration [R4].
5. Typically, primitive-based methods (like 4DGS [R4] or STG [R5]) provide better performance than deformation-based methods. The proposed method is built upon deformation-based method. It is interesting to investigate if this Laplacian decomposition works and can be integrated in other paradigms.
6. From the ablation study, it seems that the adaptive split strategy does not provide sufficient gains.

[R1] Bae, Jeongmin, Seoha Kim, Youngsik Yun, Hahyun Lee, Gun Bang, and Youngjung Uh. "Per-gaussian embedding-based deformation for deformable 3d gaussian splatting." In European Conference on Computer Vision, 2024.

[R2] Guo, Zhiyang, Wengang Zhou, Li Li, Min Wang, and Houqiang Li. "Motion-aware 3d gaussian splatting for efficient dynamic scene reconstruction." IEEE Transactions on Circuits and Systems for Video Technology (2024).

[R3] Li, Hao, Sicheng Li, Xiang Gao, Abudouaihati Batuer, Lu Yu, and Yiyi Liao. "GIFStream: 4D Gaussian-based Immersive Video with Feature Stream." In Proceedings of the Computer Vision and Pattern Recognition Conference, 2025.

[R4] Yang, Zeyu, Hongye Yang, Zijie Pan, and Li Zhang. "Real-time photorealistic dynamic scene representation and rendering with 4d gaussian splatting." arXiv preprint arXiv:2310.10642 (2023).

[R5] Li, Zhan, Zhang Chen, Zhong Li, and Yi Xu. "Spacetime gaussian feature splatting for real-time dynamic view synthesis." In Proceedings of the IEEE/CVF Conference on Computer Vision and Pattern Recognition, pp. 8508-8520. 2024.

**Questions:**

1. The authors are expected to provide clarifications on how the proposed contributions benefit the proposed pipeline comprehensively. The differences between the proposed method and baselines should be enhanced.
2. The explanations on the deformation calculation mechanism, especially how the attention calculation works, is recommended to be enhanced.
3. Consistent with Weakness 5, it is curious to think whether the proposed design is applicable to primitive-based 4DGS methods, as the Laplacian decomposition should be a general motion representation format.
4. The comparison should include more results to show the effectiveness of the proposed method. The Laplacian decomposition is an interesting part in this paper. It is suggested to include more content on verifying its effectiveness. For example, the frequency distribution can be visualized to find some insights. This is not a weakness or compulsory question for the authors.

---

### Official Review · Reviewer_cN8c · 2025-10-27

**Soundness:** 2
**Presentation:** 1
**Contribution:** 2
**Rating:** 4
**Confidence:** 5

**Summary:**

This paper rethinks dynamic 3DGS through Laplacian spectral analysis, providing a hybrid framework for local frequency analysis. Meanwhile, it focuses on the dynamic properties of each Gaussian and the optimization issues in the derivation process, proposing a novel explicit–implicit hybrid algorithmic model.

**Strengths:**

The paper proposes a novel Laplacian-based transformation to enhance dynamic modeling capability.
It also introduces an adaptive Gaussian segmentation strategy that automatically adjusts the original density and anisotropy through KDTree-guided spectral analysis.

**Weaknesses:**

1. The paper’s writing is somewhat disjointed, and several details are unclear.
Many of the arguments lack solid experimental support — for example, line 68: “existing deformable methods suffer from,” and line 211: “Incorporation of learnable frequencies,” etc.

2. Methods based on deformation fields are inherently limited in handling complex temporal variations (such as fast motion or object disappearance/reappearance). Although the authors attempt to improve this class of methods, they do not provide sufficient experimental evidence to support their claims.

3. In Fig. 4 (a) and (b), the results only demonstrate that the Laplacian motion flow is effective, but they do not show that its quality surpasses the current SOTA. Many existing methods have already achieved excellent results in this scenario.

4. Regarding dataset selection, current dynamic methods have already performed well on datasets such as D-NeRF and N3DV, where reconstruction quality is close to saturation. The real challenge now lies in large-scale dynamic reconstruction. It is strongly recommended to evaluate the proposed method on large-scale datasets such as  Nvidia Dynamic Scene datasets [1], Dynamic3DGS [2] or VRU [3]  for more convincing results. If the proposed approach can efficiently model complex dynamic scenes at this scale, it would be truly exciting. In addition, providing supplementary video results would further strengthen the paper’s credibility.


[1] Neural Trajectory Fields for Dynamic Novel View Synthesis

[2] Dynamic 3D Gaussians: Tracking by Persistent Dynamic View Synthesis

[3] Swift4D: Adaptive divide-and-conquer Gaussian Splatting for compact and efficient reconstruction of dynamic scene

**Questions:**

1. How are $A_s$ and $A_l$ combined and fed into the MLP?
2. What about the training time, inference speed, and storage requirements?
3. The ablation study is insufficient — what happens if the hash module is removed? Is the dynamic training mainly driven by the hash module?

---

### Official Review · Reviewer_Hger · 2025-10-29

**Soundness:** 3
**Presentation:** 2
**Contribution:** 1
**Rating:** 4
**Confidence:** 4

**Summary:**

This paper proposes a unified dynamic radiance field framework that integrates Laplacian spectral analysis with 3D Gaussian Splatting (3DGS). The authors claim that decomposing motion into multiple frequency components allows for more stable and detailed reconstruction of dynamic scenes. The approach also incorporates per-Gaussian dynamic attributes and an adaptive Gaussian splitting strategy to handle complex motion patterns. Experiments are conducted on Neu3DV and Hyper-NeRF datasets, showing competitive results.

However, the paper’s originality and empirical justification are limited. The idea of modeling motion in the frequency domain is not novel—similar decompositions have been explored in Shape of Motion (2024), which also represents motion as weighted combinations of frequency components. Furthermore, the proposed method does not outperform strong baselines such as Motion-Aware Gaussian Splatting, which achieves higher PSNR on both Neu3DV (33.26 dB vs 32.12 dB) and Hyper-NeRF (27.87 dB vs 25.82 dB). The lack of motion trajectory visualization and testing on high-motion datasets further weakens the empirical validation.

**Strengths:**

- The paper attempts to connect **spectral Laplacian analysis** with **explicit Gaussian primitives**, providing an interesting hybrid theoretical perspective.
- The **framework design** (spectral decomposition + per-Gaussian dynamics + adaptive splitting) is logically coherent.

**Weaknesses:**

1. **Limited Novelty**
   - The concept of decomposing motion into frequency components is not new. *Shape of Motion*[1] (2024) already investigated spectral motion representation and weighted recomposition within a Gaussian Splatting framework.

2. **Insufficient Empirical Advantage**
   - The claimed superiority in reconstruction quality is not supported by quantitative results. *Motion-Aware GS*[2] achieves higher PSNR on both Neu3DV dataset (33.26 dB) and Hyper-NeRF dataset(27.87 dB) than the proposed method (32.12 dB and 25.83 dB respectively).

3. **Lack of Motion Visualization and Analysis**
   - The paper provides only abstract schematic figures to illustrate Laplacian frequency decomposition, with **no visualization of motion trajectories, spectral components, or temporal consistency**.
   - Without explicit motion visualizations or error heatmaps, it is difficult to assess whether the frequency-domain modeling genuinely captures complex motion.

4. **Limited Dataset Diversity**
   - All evaluated datasets involve smooth or slow motion. The method is not tested on **high-motion or large-deformation datasets** such as CMU-Panoptic used in *D-3DGS*[3], Kubric, or the NVIDIA Dynamic Dataset used in *Shape of Motion*.
   - As a result, the generalization of the proposed approach to challenging real-world motion remains unclear.

5. **Unclear Advantage in Model Characteristics**
   - The claimed benefits (stability, detail, adaptability) are qualitative and not supported by consistent quantitative or visual evidence.
   - The framework increases complexity without demonstrating clear performance or interpretability gains.

6. **Weak Method Organization and Missing Formulation**
   - The **method section (Section 3)** lacks a clear structure and summary. A concise overview at the beginning would help readers grasp the relationship between components.
   - There is **no explicit formulation or algorithm** describing how the Laplacian motion field \( L(t) \) and the spatial hash features actually influence or update Gaussian attributes.
   - Subsection 3.1.2 is disproportionately long and focuses heavily on basic Laplacian decomposition concepts that are well-established in information processing. The presentation could be streamlined and redirected toward showing the actual coupling mechanism between spectral motion and Gaussian updates.


```
[1] Wang, Qianqian, et al. "Shape of motion: 4d reconstruction from a single video." Proceedings of the IEEE/CVF International Conference on Computer Vision. 2025.
[2] Guo, Zhiyang, et al. "Motion-aware 3d gaussian splatting for efficient dynamic scene reconstruction." IEEE Transactions on Circuits and Systems for Video Technology (2024).
[3] Luiten, Jonathon, et al. "Dynamic 3d gaussians: Tracking by persistent dynamic view synthesis." 2024 International Conference on 3D Vision (3DV). IEEE, 2024.
```

**Questions:**

See weaknesses.

---

### Official Review · Reviewer_wPrG · 2025-11-01

**Soundness:** 2
**Presentation:** 2
**Contribution:** 2
**Rating:** 4
**Confidence:** 4

**Summary:**

This paper proposed a dynamic reocnstruction method based on the 3DGS technology. They integrateed laplacian spectral analysis to the motion modeling to solve the problem in dynamic reconstruction like over-smoothing. Meanwhile, they combined some regularization and split strategies to improve the robustness.

**Strengths:**

1. This paper tried to inject the laplacian analysis to the dynamic reconstruction to improve the fidelity.
2. This paper introduceda KDTree-guided strategy to adaptively split Gaussians.

**Weaknesses:**

1. The adopted benchmark seems already dont have huge challengy to prove the effectiveness. Many previous methods have shown the similar effects, and I encourage the author to choose more challenging benchmarks and to find the improvement aspect.
2. The declared advantage in the motion modeling is not demonstrated clearly.
3. Maybe needs to add more discussion with existing methods that adopt the fourier analysis methods, which all use the predifined trajectory prior.

**Questions:**

see weakness.

---

### Note · Authors · 2025-11-24

I have read and agree with the venue's withdrawal policy on behalf of myself and my co-authors.